# Dietary Capsaicin: A Spicy Way to Improve Cardio-Metabolic Health?

**DOI:** 10.3390/biom12121783

**Published:** 2022-11-29

**Authors:** Arpad Szallasi

**Affiliations:** Department of Pathology and Experimental Cancer Research, Semmelweis University, 1085 Budapest, Hungary; szallasi.arpad@med.semmelweis-univ.hu

**Keywords:** capsaicin, nutraceutical, cardio-metabolic health, dyslipidemia, hyperglycemia, insulin resistance, gut microbiota, low-grade inflammation

## Abstract

Today’s sedentary lifestyle with too much food and too little exercise has made metabolic syndrome a pandemic. Metabolic syndrome is a major risk factor for type-2 diabetes and cardiovascular disease. New knowledge of medical and nutraceutical intervention in the early stages of metabolic syndrome is central to prevent these deadly complications. People who eat chili pepper on a regular basis seem to stay healthier and live longer than those who do not. Animal experiments suggest a therapeutic potential for dietary capsaicin, the active principle in hot chili pepper, to reduce the risk of developing metabolic syndrome. This is an attractive theory since capsaicin has been a culinary staple for thousands of years, and is generally deemed safe when consumed in hedonically acceptable doses. The broad expression of the capsaicin receptor TRPV1 in metabolically active tissues lends experimental support to this theory. This review critically evaluates the available experimental and clinical evidence for and against dietary capsaicin being an effective dietary means to improve cardio-metabolic health. It comes to the conclusion that although a chili pepper-rich diet is associated with a reduced risk of dying due to cardiovascular disease, dietary capsaicin has no clear effect on blood glucose or lipid profiles. Therefore, the reduced mortality risk may reflect the beneficial action of digested capsaicin on gut microbiota.

## 1. Introduction

The term “metabolic syndrome” was coined by Herman Haller in the 1977 [1], but developing a standard definition has proved challenging [2]. Indeed, metabolic syndrome is not a disease itself [3], but rather a constellation of conditions that, if left untreated, together raise the risk of coronary heart disease (CHD), sudden cardiac death, stroke, type-2 diabetes (T2DM), and non-alcoholic fatty liver disease (NAFLD) [4,5,6]. These conditions include central obesity (excess body fat around the waist), hypertension, high blood sugar, and dyslipidemia (abnormal cholesterol and triglyceride levels). The key event that initiates the development of metabolic syndrome is now thought to be central obesity [7,8]. With almost 40% of adults worldwide being overweight or obese, obesity has already reached pandemic proportions [9]. It is generally agreed that lifestyle changes (little exercise and increased access to unhealthy fast food) are to blame for this trend [10,11].

According to the National Cholesterol Education Program (NCEP) definition, roughly one in three American adults have metabolic syndrome [12]. Patients with metabolic syndrome have an increased probability (hazard ratio, HR) of dying of CHD or other cardiovascular disease (CVD) [13,14]. Therefore, both from clinical and public health perspective, it is of utmost importance to diagnose and treat metabolic syndrome in its early stages before the development of T2DM and CVD. The prescription should include lifestyle intervention (a combination of exercise and healthy diet) [15] and, if necessary, pharmacotherapy. Since hyperglycemia and dyslipidemia seem to be the main drivers of adverse outcomes [16,17], pharmacotherapy is aimed at returning blood glucose and lipid levels into their normal range. These medications are, however, not without side-effects; therefore, nutraceuticals (food ingredients that may provide health benefits in addition to their nutritional value) are attracting attention as natural remedies [18,19,20].

Hot chili pepper is eaten on a daily basis by an estimated quarter of the world’s population. Large, population-based epidemiological studies suggest that chili-eaters may live longer and stay healthier than those who prefer bland food [21,22,23]. In fact, chili-eaters were found to have a reduced hazard ratio of 0.82 for dying due to CVD [24]. It was proposed that chili pepper may be the long-sought after answer to the “how to live longer” question [25].

The main pungent ingredient in hot chili peppers is capsaicin. Connoisseurs of hot spicy food know the predominant pharmacological actions of capsaicin from personal experience. The hot, burning sensation in the mouth is followed by a refractory state, traditionally termed “desensitization” [26,27,28,29]. Capsaicin also affects thermoregulation (known as “gustatory sweating” [30]) that may explain the popularity of hot pepper under tropical climates [31]. Importantly, capsaicin consumption is generally deemed safe at hedonically acceptable doses.

So, to stay healthy, pass me the pepper shaker please? Should we recommend that chili pepper (or pure capsaicin) be introduced into the diet of people with metabolic syndrome? Although animal experiments seem to support this hypothesis, human observations are far from being conclusive. This review aims to critically evaluate the evidence for and against capsaicin as an effective nutraceutical to prevent and/or treat cardio-metabolic syndrome.

## 2. Dietary Capsaicin in Animal Experiments: Effects on Blood Glucose and Lipid Profile

A large body of literature has shown the influence of dietary capsaicin on blood glucose levels in experimental animals. Didactically, the literature can be divided into three major groups: (1) healthy animals on regular or high-fat diet (HFD); (2) animal models of obesity, and (3) animal models of diabetes.

In in vitro organ perfusion studies, capsaicin reduced glucose absorption from the small intestine of the rat [32,33]. Yet, intragastric administration of habanero pepper (0.025 g dry matter daily for 28 days) had no measurable effect on blood glucose levels [34].

In non-obese mice on high-fat diet (HFD), increased glucagon-like peptide-1 (GLP-1) secretion was observed after 8 weeks of capsaicin supplementation [35,36]. With regard to the mechanism of action of capsaicin, the authors of these two studies reached the opposite conclusion: the first study implicated the capsaicin receptor transient receptor potential, vanilloid-1 (TRPV1) since the capsaicin effect on GLP-1 was absent in *Trpv1* null mice [35]; the second study, however, described an off-target capsaicin action on gut microbiota [36]. Capsaicin effects on gut microbiota are detailed later.

In C57BL/6J mice on HFD (32% fat), dietary capsaicin (0.01% in chow) co-administered with eicosapentaenoic acid lowered fasting glucose, insulin, and triglyceride levels compared to placebo [37].

In male Wistar rats (8–9 weeks old, weighing, on average, 338 g) fed high-carbohydrate/HFD, dietary capsaicin (0.015%, corresponding to 7.3 mg/kg/day) significantly reduced daily calorie intake with corresponding reduction in weight gain and visceral adiposity (Table 1) [38]. These changes were accompanied by decreased total-cholesterol levels, but no change in blood glucose [38]. In male Sprague-Dawley rats on HFD, 0.02% capsaicin supplemented to the diet for 4 weeks reduced the rise in serum lipids, increased the expression of the insulin receptor in the muscle, and lowered plasma leptin levels, indicating a coordinated capsaicin effect on energy metabolism [39].

In *ob/ob* mice (a genetic model of obesity), capsaicin (0.02% for 6 weeks) attenuated fasting glucose levels compared to placebo [40]. In KK-A(y) obese/diabetic mice, supplementation of the diet with 0.0042% capsaicin and 0.031% caffeine resulted in an attenuated rise in blood glucose (from to 132 mg/dL to 255 mg/dL) compared to placebo (from 234 mg/dL to 500.6 mg/dL) during the 28-day course of the study [41]. In a second study with KK-A(y) mice on HFD, dietary capsaicin (0.015%) markedly decreased fasting triglyceride levels, blocked macrophage infiltration into the adipose tissue, and suppressed the expression of the pro-inflammatory adipokine genes [42].

In spontaneously hypertensive rats fed a regular diet, dietary capsaicin (0.5 or 1 mg/kg) given for 12 weeks between 17 and 30 weeks of age produced no difference in serum glucose and cholesterol levels (Table 2) [43].

In streptozotocin (STZ)-induced diabetic rats (a model of type-1 diabetes), dietary capsaicin (6 mg/kg for 28 days) blocked the intestinal absorption of glucose, resulting in significantly lower blood glucose levels in the capsaicin group than in the control group: 14.7 mM and 19.3 mM, respectively [44,45]. In pacreatectomized diabetic rats (another model of type-1 diabetes), capsaicin reduced visceral fat accumulation and improved glucose tolerance, though, somewhat unexpectedly, the non-pungent capsaicin congener, capsiate, was more effective than capsaicin in this study [46].

In dogs, dietary capsaicin reduced fasting blood glucose from 6.4 mmol/L to 4.91 mmol/L compared to placebo [47].

In rats, capsaicin blocked hepatic fat accumulation [38,48]. Moreover, dietary capsaicin lowered serum triglyceride levels both in rats [49] and guinea pigs [50] on HFD. Importantly, in guinea pigs capsaicin (2.5–10 mg/kg per day for 1 month) also reduced atherosclerotic plaque formation by 17.9% [51].

Unexpectedly, capsaicin (4 mg/animal/day) reduced triglyceride, total cholesterol and LDL-cholesterol levels in turkey poults fed 0.2% cholesterol [52]. This is perplexing since birds are unresponsive to capsaicin [53] due to a mutation in the capsaicin recognition domain of the TRPV1 receptor [54,55].

## 3. Eating Chili Pepper Makes You Healthy?

Three large epidemiological studies have reported health benefits for chili-eaters. The National Health and Nutritional Examination Survey [21] followed 16,179 Americans for 6 years (from 1988 to 1994) and found that the mortality rate among chili-eaters (21.6%) was significantly lower than in non-eaters (33.6%). The prevalence of hypertension was lower in the chili-eater group (19.9%) than among non-eaters (27.1%), which may explain the reduced hazard ratio (0.86) of dying from heart attack or stroke. However, no significant difference in total cholesterol (202 mg/dL versus 205 mg/dL) or HDL-cholesterol (50 mg/dL versus 51 mg/dL) levels were detected between chili-eaters and non-eaters, respectively. In accordance, a recent umbrella review of 11 systematic reviews and meta-analyses found no significant association between spicy food intake and blood glucose, plasma insulin, and hemoglobin A1_c_ levels [56].

The China Kadoorie Biobank study prospectively followed 199,293 men and 288,082 women aged 30 to 79 years for a median of 7.2 years [22]. The consumption of spicy food was self-reported. Participant who ate spicy food every day showed a 14% relative risk reduction in mortality compared to those who consumed spicy food less than once a week. Spicy food consumption was inversely associated with the risk of death to due ischemic heart disease.

An Italian study (Moli-sani Study cohort, 22,811 subjects followed from 2005 to 2010) found a 0.56 HR risk of dying to ischemic heart disease in the chili-eater group [23].

Recent analysis of data pooled from four large epidemiological studies (US, China, Italy and Iran) involving a total of 564,748 individuals came to similar conclusion: regular consumers of spicy food (at least once a day) experienced a significant decrease in heart-disease related mortality (HR/RR, 0.82) compared to non-eaters (none, or less than 1 spicy meal a week) [24].

From these studies one may conclude that dietary capsaicin protects against the most lethal complications of metabolic syndrome, acute myocardial infarction and stroke. However, is this true, or are we jumping into this conclusion prematurely? To answer this, first one has to ask the question why some people eat hot pepper on a daily basis whereas others never do.

Eating hot pepper is an acquired taste, and as such a quite perplexing one. During evolution, plants have developed capsaicin as a chemical self-protecting weapon to deter herbivores [57]. Indeed, foresters use capsaicin to protect the bark of young trees from deer [58,59], and bird fanciers add capsaicin to bird feed to repel rodents [60]. This works because birds have lost their capsaicin sensitivity due to mutations in the capsaicin receptor TRPV1 [54,55]. Thus, birds can eat chili pods and spread the pepper seeds in the feces [57].

Several competing theories have been formulated to explain why the same hot taste that repels most animals is found pleasurable by many people. A popular theory posits that capsaicin exerts a pleasant cooling effect after meal by evoking gustatory sweating (capsaicin as “natural air conditioner”) [30,31]. This may be true, but it still does not explain why many people who live under tropical climates dislike capsaicin, nor does it explain why inhabitants of cold countries enjoy eating hot spicy food [61]. So maybe eating hot pepper really is a personality trait [62,63] (sort of “culinary masochism” [64]) that has nothing to do with the thermoregulatory actions of capsaicin.

If we don’t really understand why people like or dislike capsaicin, we cannot determine the differences, if any, between the two groups either. Maybe chili-eaters are more active physically. Or maybe chili pepper is a staple of a healthier diet. Indeed, authors of the Chinese epidemiological study point out that regular chili-eaters tend to be more rural, and mostly peasants or manual laborers [22]. These people are probably more active physically and have less access to unhealthy processed food than their compatriots who live in the cities.

In conclusion, although enjoying hot spicy food seems to be associated with reduced morbidity and mortality, it is unclear from these epidemiological studies if this is due to capsaicin or not. This question can be better addressed in controlled clinical trials.

## 4. Clinical Studies with Dietary Capsaicin: Effect on Blood Glucose

In a clinical study with 42 healthy, non-obese subjects with low HDL-cholesterol, dietary capsaicin (4 mg per diem for 3 month) had no measurable effect on blood glucose levels [65]. Similarly, a meta-analysis of 14 human trials with long-term dietary capsaicin supplementation found no effect on fasting blood glucose compared to placebo [66]. Capsaicin had no effect on 2 h post-prandial blood glucose either.

In obese individuals (waist circumference > 94 cm), chili pepper (African bird's eye chili) consumption (5.82 mg capsaicin per meal) showed a tendency for lowering blood glucose, but the observed changes did not reach significance [67].

In the oral glucose tolerance test (OGTT) administered to 12 healthy volunteers, per os capsaicin was reported to increase plasma insulin levels, resulting in attenuated blood glucose [68]. Another study, however, reported increased glucose absorption (with corresponding increase in blood glucose) in 14 healthy volunteers in the glucose loading test (75 g glucose) following capsaicin administration [69].

In a placebo-controlled, blinded, cross-over experiment, eight young adult male participants (average age 22 years) were fed a high carbohydrate meal (90 g glucose) after overnight fasting: this increased blood glucose to 8.5 mmol/L from a fasting baseline of 4.4 mg/L 45 min after the meal [70]. At this point, the volunteers consumed *Capsicum annum* powder (475 mg) or placebo. Blood glucose returned to the normal range within 15 min of capsicum supplementation. In the placebo group, it took 120 min for the blood glucose to normalize.

Finally, in 44 pregnant women with gestational diabetes, capsaicin (5 mg per day for 4 weeks) reduced 2 h post-prandial glucose and reduced the incidence of large for gestational age (LGA) infants [71].

## 5. Clinical Studies with Dietary Capsaicin: Effect on Serum Lipids

In a randomized cross-over study, twenty-seven volunteers consumed freshly chopped chili pepper (30 g/day) for 4 weeks. No difference in serum lipid levels were noted compared to those who ate bland food at the end of the dietary period [72]. However, the rate of lipid oxidation was significantly lower after the chili diet [72]. In another study that employed 42 subjects with low HDL-cholesterol, dietary capsaicin (4 mg daily for 3 months) was associated with a very modest, non-significant increase in fasting HDL-cholesterol (0.92 mM, compared to 1 mM in the placebo group) [65].

Capsimax is a concentrated red chili pepper extract that contains 2 mg capsaicin, and is marketed as nutritional supplement (“weight loss pill”) to “burn calories” [73]. Seventy-seven healthy volunteers took two capsules of Capsimax a day for 12 weeks, resulting in a mild reduction in serum HDL-cholesterol, but no change in blood glucose [74].

A meta-analysis of 10 controlled clinical trials with 398 participants also found decreased total cholesterol levels in the capsaicin groups compared to placebo [75]. Of note, the change in total cholesterol was very modest (from 160 to 152 mg/dL) and remained within the normal range (125–200 mg/dL) [75].

## 6. Dietary Capsaicin and Metabolic Health: Possible Mechanisms of Action

Although the data are conflicting, for the sake of argument let us accept that dietary capsaicin exerts a beneficial effect on blood glucose and serum lipids. Then one has to ask the following questions:

(1) Are these effects on-target (that is, mediated by the capsaicin receptor TRPV1) or off-target?

(2) If on-target, what are the major TRPV1-expressing players in blood glucose and lipid regulation?

The capsaicin receptor TRPV1 [76] is highly expressed on unmyelinated sensory nerves [26,27,28,29]. It is also expressed, albeit at much lower levels, in metabolically active tissue, including adipocytes [77], skeletal muscle cells [78], pancreatic beta-cells [79], and hepatocytes [80]. Since TRPV1-positive nerves innervate the Langerhans islets in the pancreas where the TRPV1-expressing beta cells reside [81,82], capsaicin may exert a complex effect on insulin secretion.

Global *Trpv1* gene inactivation by genetic manipulation (*Trpv1* null animals) probably eliminates the TRPV1 protein from all cells. By contrast, systemic capsaicin treatment is likely to deplete (or “defunctionalize”) the TRPV1-expressing nerves only [26,27,28,29], leaving non-neuronal TRPV1 expression intact. Adult capsaicin treatment is preferable since animals may develop compensatory mechanisms following neonatal capsaicin treatment.

The phenotype of the *Trpv1* null mouse is intriguing. These animals are hyperactive and lean when young, and they become lazy and overweight when they grow old [83]. Glucose-induced insulin secretion is blunted in *Trpv1* null mice kept on a normal diet [84]. When kept on HFD, *Trpv1* null animals showed a 40% reduction in glucose metabolism and grow even more obese and insulin resistant than the wild-type controls [85]. Dietary capsaicin stimulates insulin and GLP-1 secretion in the wild-type mice but not the KO mice, implying a TRPV-mediated action [35]. In another study, however, the effect of capsaicin on GLP-1 secretion was transferable with fecal transplant [36], indicating an off-target capsaicin action on the gut flora.

In healthy adult rats, the effect of desensitization by systemic capsaicin administration on glucose tolerance is yet to be investigated. More is known about the glucose homeostasis of normal or diabetic rats whose sensory afferents had been ablated by capsaicin treatment as neonates. Ablation of sensitive-afferents by neonatal capsaicin administration improved the oral glucose tolerance in normal and STZ diabetic rats, with no measurable effect on insulin secretion [86]. In non-diabetic but obese Zucker rats, capsaicin desensitization improved glucose tolerance [87]. Similarly, in diabetic Zucker fatty rats, ablation of TRPV1-expressing sensory nerves by capsaicin, or its ultrapotent analog, resiniferatoxin, increased insulin secretion, leading to improved glucose tolerance [82,88].

In adult Wistar rats whose TRPV1-expressing nerves were eliminated by neonatal capsaicin administration, no difference in blood glucose was found in the i.v. glucose tolerance test [89]. These animals, however, displayed much attenuated glucose-induced insulin release [89]. Likewise, blunted glucose-induced insulin secretion was observed in the *Trpv1* null animals [35]. Furthermore, low-dose GLP-1 augments insulin response in control mice, but not after neonatal capsaicin administration [90]. These results were interpreted to imply that capsaicin-sensitive nerves are involved in insulin sensitivity, but not blood glucose regulation.

In conclusion, the role of capsaicin-sensitive afferents in physiological glucose homeostasis is poorly understood. In animal models of type-2 diabetes, defunctionalization of these nerves seem to improve glucose tolerance.

## 7. Can Capsaicin Prevent or Ameliorate Metabolic Syndrome?

Most authorities agree that central obesity is the defining factor in metabolic syndrome [7,8]. Obesity is associated with insulin resistance, a major component in metabolic syndrome, through chronic low-grade inflammation [91,92,93]. Eventually, insulin resistance progresses into type-2 diabetes [94]. Capsaicin may prevent the development of central obesity and its complications via a complex and poorly understood mechanism of action (Figure 1) [95].

Capsaicin-sensitive nerves are thought to play an important role in the low-grade chronic inflammatory reaction of obesity [96]. Indeed, in animal models of type-2 diabetes, ablation of these nerves by capsaicin or resiniferatoxin (an ultrapotent capsaicin analogue) improves blood glucose [82,87,88]. The roles of capsaicin-sensitive nerves in obesity [97] and diabetes [96,98] have been detailed elsewhere. Here it suffices to mention that chronic low-grade inflammation increases the production of reactive oxygen species (ROS) in the adipose tissue that, in turn, disturbs adipokine production [99,100,101]. Adipokines (such as leptin, plasminogen activator inhibitor-1, PAI-1, and monocyte chemoattractant protein-1, MCP1) are cell signaling proteins secreted by the adipose tissue with key roles in metabolic syndrome. For example, PAI-1 is present in increased levels during obesity and metabolic syndrome [102]. Furthermore, MCP1 impairs insulin signaling in skeletal muscle, thereby contributing to insulin resistance [103].

TRPV1-expressing nerves are a major source of calcitonin gene-related peptide (CGRP) [26,27,28]. In Zucker rats, increasing plasma CGRP heralds the development of obesity [87]. In these animals, ablation by capsaicin of the TRPV1-expressing nerves prevents the increase in plasma CGRP and reduces the fasting glucose from 5.1 mmol/L to 4.3 mmol/L [85]. In obese women, plasma CGRP was significantly higher than in the control group: 32.26 pg/mL and 21.64 pg/mL, respectively [104]. Gaining weight in the elderly has also been linked to increasing CGRP levels [96].

In theory, dietary capsaicin can block CGRP release by desensitizing TRPV1-positive nerves. Since elevated plasma CGRP is believed to play an important role in the development of metabolic syndrome, capsaicin should prevent, or at least slow, the progression of metabolic syndrome. Here the key question is whether serum capsaicin levels can reach concentrations high enough to desensitize neuronal TRPV1.

In rats, approximately 25% of the digested capsaicin is absorbed from the intestine [105]. The absorbed capsaicin, however, undergoes extensive metabolism in the liver [106,107]. Thus, the actual plasma concentration of capsaicin is probably fairly low. Indeed, only 5% of the intragastrically administered [^3^H]dihydrocapsaicin was detected in the blood 15 min after application [108]. This is all the better, since capsaicin injected i.v. may provoke chest pain, cough, hypotension, bradycardia, and, eventually, cardiovascular collapse [109,110,111,112]. Even mucosal exposure to high-dose capsaicin (for instance, “pepper spray”, used in law enforcement, crowd control, and personal protection) may have fatal consequences [113,114,115].

In conclusion, dietary capsaicin can topically desensitize nerves endings in the mouth, but it is unlikely to ameliorate CGRP release systemically.

## 8. Capsaicin, Gut Microbiota, and Metabolic Syndrome

“We are what we eat” is true in more than one sense: what we eat will either help maintain a healthy gut microbiota or cause disease by creating ground for the overgrowth of pathogens. As indicated by the high number of reviews in PubMed (787), there is a very large body of experimental evidence linking abnormal gut microbiota (so-called “dysbacteriosis”) to metabolic syndrome [116,117,118,119]. The gut-centric theory of metabolic syndrome posits that an unhealthy diet (for example a diet rich in fat and poor in fibers) permits the growth of pathogens in the colon, leading to metabolic endotoxemia and impaired insulin sensitivity [116].

Humans lack the biochemical machinery needed to break down complex carbohydrates—the gut bacteria do this work for us. The bacterial fermentation product of these complex carbohydrates includes a mixture of butyrate, propionate, and acetate. Butyrate and propionate are good; acetate, however, may be bad. Butyrate promotes the health of enterocytes in the intestine [120], whereas propionate suppresses appetite by releasing the satiety hormone, peptide YY [121]. By contrast, acetate may cause hyperphagia, and ultimately obesity, by activating the parasympathetic nervous system [122]. Indeed, obese people have higher acetate levels in their stool than lean individuals [123]. The delicate balance between propionate- and acetate-producing bacteria may help maintain a healthy body weight. Conversely, the overgrowth of acetate-producing bacteria may be an important factor in obesity [124].

As discussed above, obesity is the key event in the development of metabolic syndrome [7,8]. Strong evidence links obesity to altered colonic microbiota [125]. Obese men show increased intestinal *Firmicutes*:*Bacteroidetes* ratio compared to people with normal body weight [126]. Experimentally, obesity is transmissible by fecal microbiota transplant [127,128]. That is, gut bacteria harvested from fat people make lean mice fat. Conversely, mice could be rescued from fecal transplant-induced obesity by the microbiota of lean animals [127]. Fecal microbiota transplant is being tested as a novel biotherapeutic intervention in obesity and diabetes [129]. Clinical trials are already on-going [130], and the initial results are promising. For example, in patients with severe obesity and metabolic syndrome, fecal microbial transplantation from lean donors improved insulin sensitivity with no serious adverse effects [131].

Diet is the main influencer of gut microbiota. Food additives, such as capsaicin, can promote the growth of “healthy” or, alternatively, pathogenic gut bacteria (Figure 1) [132,133,134]. As discussed above, only 25% of the digested capsaicin is absorbed from the intestine [105]; therefore, capsaicin can reach high concentrations in the feces. As the Hungarian saying goes, “hot pepper bites twice”. Capsaicin can affect gut microbiota both directly and indirectly. The antibacterial action of capsaicin is well-documented [135,136,137]. To explain the popularity of chili pepper in countries where food poisoning has been a problem historically, it was speculated that capsaicin keeps the food safe by preventing the growth of harmful bacteria (“capsaicin as a natural refrigerator”) [138]. In addition, capsaicin can stimulate mucin production in the colon [139]. This is important because mucin supports the growth of “good” bacteria such as the “anti-obesity” bacterium, *Akkermansia muciniphila* (Figure 1) [140].

In the stool of rats fed HFD, the pro-inflammatory bacterium *Bilophila wadsworthia* thrived [141], whereas the anti-obesity bacterium, *Akkermansia muciniphila,* declined [142]. In mice fed HFD, dietary capsaicin (2 mg/kg for 12 weeks) reversed this pathogenic trend: it stimulated the growth of *Akkermansia mucinophila*, and other propionate-producing bacteria [143,144]. Of note, the beneficial action of capsaicin was noted both in *Trpv1* null and wild-type mice [144], implying a non-TRPV1-mediated action.

Chronic low-grade inflammation is thought to be a defining event in the pathogenesis obesity and insulin resistance [91,92,93,96]. This inflammatory reaction may be also intimately related to the metabolic defects seen in metabolic syndrome. Several lines of experimental evidence support this hypothesis. In experimental animals, overexpression of tumor necrosis factor-α (TNFα) leads to insulin resistance [99]. Conversely, mice lacking TNFα due to targeted mutation are protected from obesity-induced insulin resistance [142]. In accord, elevated TNFα mRNA levels were found in human obese adipose tissue [145,146].

An overgrowth of the *Muribaculaceae* (previously known as S24-7) bacterium family in the stool is associated with colitis [147]. Moreover, lipopolysaccharide (LPS)-producing gut bacteria can cause metabolic endotoxemia that, in turn, may help maintain the low-grade inflammation in the liver and adipose tissue [148]. Capsaicin may represent a novel nutraceutical strategy to prevent this endotoxemia and the resultant chronic inflammation. Indeed, capsaicin was reported to reduce the number of gram-negative LPS-producing bacteria in the stool [134]. Moreover, in obese mice, dietary capsaicin suppressed adipokine gene expression, and interleukin-6 (IL-6) and MCP1 were down-regulated [149].

## 9. Capsaicin and Cardiovascular Disease

There is an obligate role for gut bacteria to convert dietary phosphatidylcholin into pro-atherosclerotic trimethylamine N-oxide (TMAO) [150,151,152]. Indeed, in men serum TMAO levels correlate with atherosclerosis [152].

Dietary capsaicin was shown to mitigate the development of atherosclerotic plaques in several species (mouse, rat, guinea pig, and hamster) [153,154,155]. In guinea pigs fed HFD, per os capsaicin reduced atherosclerotic plaque area by 18% compared to placebo [51]. This beneficial effect was attributed to the increased superoxide dismutase (SOD) activity and nitric oxide (NO) production [150]. In hamsters, dietary capsaicin (1.3 mmol) reduced the total serum cholesterol and diminished the formation of atherosclerotic plaques [154], and in *Apo-E* gene null mice fed cholesterol-rich HFD, capsaicin reduced plaque formation via the peroxisome proliferator-activated receptor-γ/liver X receptor-α (PPARγ/LXRα) pathway [155].

Capsaicin probably interferes with changes that cause atherosclerosis by both on-target (that is, TRPV1-mediated) and off-target mechanisms. TRPV1 is expressed both in vascular endothelium [156,157,158] and smooth muscle cells [159,160,161]. Activation of the endothelial TRPV1 receptor leads to an increase in NO production [157,160] that, in turn, prevents hypertension [161]. In *Apo-E* null animals, intrathecal capsaicin or resiniferatoxin suppress lipid accumulation in the vasculature [162], whereas this effect is absent in the *Apo-E/Trpv1* double knock-out animals [162], implying an on-target, TRPV1-mediated action.

Capsaicin, however, also has a well-documented antioxidant action not mediated by TRPV1 [163,164,165,166]. For example, in erythrocytes subjected to oxidative stress, capsaicin (10 μM) inhibits lipid peroxidation [166].

In addition to its effect in atherosclerotic plaque formation, capsaicin exerts a complex action on the heart. TRPV1 is expressed in the myocardium [167] with a reduced expression in the diabetic heart [168]. The biological function of myocardial TRPV1 is unknown. More is known about the function of TRPV1-expressent afferents innervating the heart. For example, these afferents regulate myocardial NO production and cGMP signaling [169]. The activation of these nerves protects the heart during ischemia [170,171]. Conversely, inactivation of TRPV1 by genetic manipulation or high-dose capsaicin administration exacerbates the myocardial damage during ischemia and impairs post-ischemic recovery [172,173,174]. In twelve patients with stable angina, a transdermal capsaicin patch was found to improve ischemic threshold [175]. In one patient, however, the patch provoked acute myocardial infarction [176,177].

## 10. Conclusions

There is a large body of experimental evidence linking the capsaicin receptor TRPV1 to metabolic syndrome [38,178,179]. TRPV1 is broadly expressed in metabolic tissues (adipose tissue, skeletal muscle, liver, pancreatic beta-cells) [77,78,79] though the biological function of this non-neuronal expression pattern remains largely unknown. More is known about the function of TRPV1-expressing sensory nerves. For example, these afferents play an important role in maintaining the low-grade chronic inflammation that characterizes metabolic syndrome [91,92,93,96,98]. TRPV1-positive nerves are also a major source of CGRP [26,27,28,29]. In Zucker rats, increasing plasma CGRP heralds the development of obesity [87]. In these animals, ablation by capsaicin of the TRPV1-expressing nerves prevents the increase in plasma CGRP and returns the fasting glucose to its normal range [87]. In obese women, plasma CGRP was significantly higher than in the control group [104].

Animals with non-functioning TRPV1-positive nerves (ablated by high-dose systemic capsaicin administration) show blunted insulin secretion in response to glucose [89], a key feature of insulin resistance. Somewhat controversially, capsaicin desensitization improves glucose tolerance in experimental models of T2DM [82,87]. Taken together, these observations imply a complex, multifaceted and poorly understood role for TRPV1 in the pathogenesis of metabolic syndrome. It has been speculated that dietary capsaicin could improve metabolic health by activating and/or desensitizing TRPV1 receptors in metabolically active tissue [177]. Indeed, three large epidemiological studies from different countries (USA, China and Italy) reported reduced cardiovascular disease morbidity and mortality in regular chili-eaters compared to non-eaters [21,22,23]; however, there is no convincing proof that dietary capsaicin can normalize blood glucose and/or prevent dyslipidemia. Dietary capsaicin undergoes rapid metabolism in the liver [106,107]; therefore, serum capsaicin is unlikely to reach concentrations sufficient to activate TRPV1 in metabolically active tissues. Indeed, in the rat, only 5% of the digested capsaicin shows up in the circulation [108]. The concentration of bioavailable capsaicin is probably even lower since serum albumin was shown to reduce capsaicin effects [180].

Capsaicin, however, may reach high concentrations in the stool to correct dysbacteriosis [132,133,134,135,143,144], acting as a probiotic nutraceutical. Since dysbacteriosis, and resultant LPS endotoxemia [148], may help maintain the low-grade chronic inflammation that characterizes metabolic syndrome [81,82,83,84,85,86,87,88,89,90,91,92,93,96], dietary capsaicin could indirectly exert an anti-inflammatory action by restoring a healthy gut microbiota.

In conclusion, chili-eaters may live longer and stay healthier that non-eaters. This beneficial effect is unlikely to be mediated by TRPV1 receptors, bur is rather related to improved colonic health. Chili pepper can be safely enjoyed in restaurant-like amounts. The capsicum capsules marketed as weight loss pills, however, should be taken cautiously and in moderation. At least one sudden cardiac death has been reported in the literature after taking cayenne pepper pills in a 41-year-old man with no prior history of cardiovascular disease [180].

## 11. Future Directions

Small molecule TRPV1 antagonists are potential analgesic, antitussive, and antiphlogistic drugs [181,182,183]. A number of TRPV1 antagonists have already entered Phase-3 clinical trials for these clinical indications [180]. TRPV1-expressing sensory nerves are also thought to participate in the low-grade chronic inflammation that characterizes diabetes and metabolic syndrome [96,98]. Indeed, in animal experiments, the TRPV1 antagonist BCTC prevented a rise in blood glucose in mice [184]. This implies a therapeutic potential for TRPV1 antagonists as novel antidiabetic agents [96,98,185]. In fact, clinical trials with TRPV1 antagonists in patients with T2DM are already ongoing [186]. Therefore, one can argue that TRPV1 antagonists may also have a clinical value in preventing or mitigating the cardiovascular complications of metabolic syndrome. This hypothesis is yet to be tested clinically.

## Figures and Tables

**Figure 1 biomolecules-12-01783-f001:**
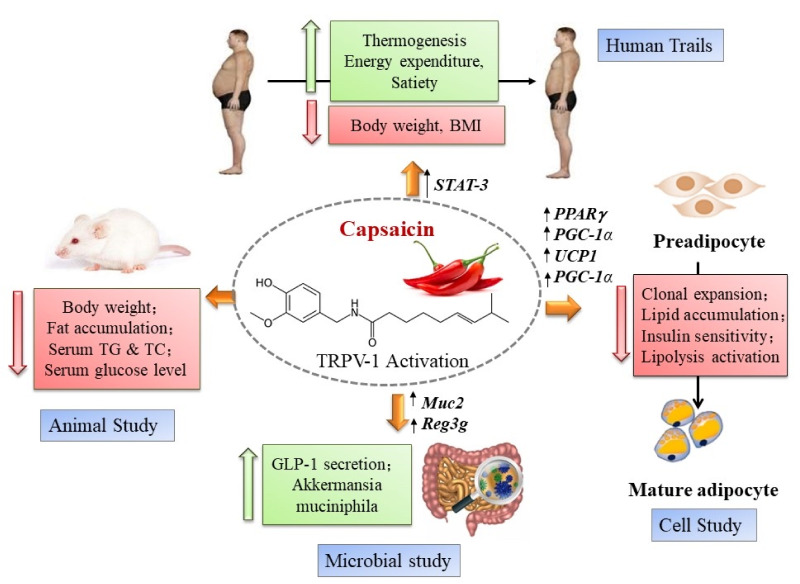
The complex molecular mechanism of action by which capsaicin may prevent the development of metabolic syndrome. In humans, TRPV1 activation may up-regulate Signal Transducer and Activator of Transcription-3 (STAT-3), a key member of the JAK/STAT pathway. Stimulation by cytokines of STAT-3 expressed in hepatocytes prevents steatosis. In animal studies, TRPV1 activation was linked to reduced fat accumulation and improved serum triglyceride (TG) and total cholesterol (TC) levels. Dietary capsaicin stimulates glucagon-like peptide-1 (GLP-1) secretion in the gastrointestinal tract. Furthermore, capsaicin supports the growth of the “anti-obesity bacterium”, *Akkermansia muciniphila*, by increasing mucin production, acting on the mucin-2 (*Muc2*) gene Activation of TRPV1 in preadipocytes results in lipid accumulation and increased insulin sensitivity via up-regulation of the Peroxisome Proliferator-Activated Receptor-γ (*PPARγ*), PPARγ-coactivator-1α (*PGC-1α*), and Uncoupling Protein-1 (*UCP1*) genes. Reproduced with permission from [95].

**Table 1 biomolecules-12-01783-t001:** Effect of dietary capsaicin (7.3 mg/kg body weight/day) administered for 16 weeks on body composition changes induced by high-carbohydrate/high-fat diet. Data are from [38].

Heading	High-Carbohydrate/Fat Diet (HCD)	HCD + Capsaicin (7.3 mg/kg/day)	*p* Value
energy intake, kJ/day	725 + 17	584 + 14	<0.0001
total fat mass, g	170.5 + 17.9	126.5 + 11.6	0.0688
BMI, g/cm^2^	0.86 + 0.02	0.73 + 0.01	<0.0001
weight gain, g	276 + 36	186 + 9	0.0056
abdominal circumference, cm	24.3 + 0.5	19.0 + 0.2	<0.0001
visceral fat, %	11.33+ 0.48	8.42 + 0.46	<0.0001

**Table 2 biomolecules-12-01783-t002:** Metabolic parameters and mean arterial blood pressure in rats on regular diet as determined after 14 days and dietary capsaicin (1 mg/kg/day) supplementation. Data are from [43].

Heading	Regular Diet	Capsaicin	*p* Value
food intake, g/24 h	19.9 + 4.7	22.7 + 7.6	0.494
total cholesterol, mmol/L	1.4 + 0.5	1.2 + 0.5	0.433
LDL-cholesterol, mmol/L	0.6 + 0.5	0.5 + 0.3	0.435
HDL-cholesterol, mmol/L	0.5 + 0.3	0.5 + 0.2	0.179
mean arterial pressure, mmHg	152 + 28	148 + 26	0.123

## Data Availability

This review is based on public databases (PubMed, etc.).

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
