# Peer review of "Dietary Capsaicin: A Spicy Way to Improve Cardio-Metabolic Health?"

_biomolecules, 2022, doi:10.3390/biom12121783_

Round 1

Reviewer 1 Report

A review by A. Szallasi concerns the effect of dietary capsaicin on cardio-metabolic health. Capsaicin is of great interest of scientist for many years, with effects on many physiological processes. Apart from its effect on thermoregulation, capsaicin has a potential as a substance of analgesic or anticancer properties. This review is focused on dietary capsaicin as a means to improve metabolism, blood glucose and lipid profiles. Author has described the available literature in detail. The possible mechanisms of capsaicin action on metabolic heath are described. Moreover, new conclusions are made about beneficial action of digested capsaicin on gut microbiota. However, author suggested this theory earlier in paper: Capsaicin for Weight Control: “Exercise in a Pill” (or Just Another Fad)? Pharmaceuticals 2022, 15(7), 851; https://doi.org/10.3390/ph15070851. To conclude, this paper is very interesting and may provide new absorbing information for readers.

Few minor suggestions:

Line 39

‘increased hazard ratio of dying in CHD’ - Please correct the sentence. Stylistically incorrect.

Line 148

0.56 HR risk of dying’ – what is a HR? – hazard ratio?

Line 552

Position 38 in literature

 https://www.eprints/usq.edu.au/33742 this page does not work. Please check it.

Line 859

Position 185 from literature is not cited in the main text

Author Response

Dear Referee: thank you for your suggestions that I fully adopted (for easier identification, all changes are highlighted in yellow in the revised MS).

1 (line 39): The sentence is now stylistically corrected.

2 (148): "HR" is now defined in line 39 where "hazard ratio" first occurs (see above).

3 reference 38 has been corrected

4 reference 185 (which is now 186 after adding an additional reference) is now cited in the main text

Reviewer 2 Report

The review article by Arpad Szallasi discusses the literature on the effects of dietary capsaicin / chili peppers on cardio-metabolic health. The review is timely, thorough, well written and discusses an important topic. I only have some minor comments the author may consider addressing.

Table 1 (ref38). It seems rats eat more in the control, compared to the capsaicin group, based on energy intake. In table 2 they seem to eat the same amount. Is that because of the different amount of capsaicin in the food 7.3 mg/kg vs 1 mg / kg ? I would assume capsaicin is aversive for rats, which may explain the difference at higher capsaicin consumption. The author may consider commenting on this and in general on whether simply eating less form the capsaicin-containing food in some of the animal studies can explain some of the effects of capsaicin .

For assessing the effect of capsaicin on human health, it would be important to know how much capsaicin enters the systemic circulation, and whether capsaicin binds plasma albumin, the most important factor being whether capsaicin can reach tissues other than the intestines in meaningful concentrations. The author discusses this to some extent, and concludes that the site of action is the intestinal microbiome, but some more quantitative description would be useful. My intuitive sense is that eating even large quantities of chili peppers does not result in tissue capsaicin concentrations that can activate TRPV1, otherwise eating chili peppers would cause pain not only at the entry and exit points, but also everywhere else.

Author Response

Dear Reviewer: thank you for your thoughful comments. For easier identification, changes are highlighted in yellow in the revised MS.

Yes, I fully agree with you, it is very difficult, if not impossible, to "blind" capsaicin even in animal experiments. Aversion to capsaicin can clearly be a factor in short-term studies, before some degree of desensitization develops. However, Table 1 summarizes results from a 16 weeks (over 100 days) experiment. I would think that rats have gotten used to (got "desensitized") to capsaicin during this long-term study. Moreover, in one study "spicy" capsaicin and "sweet" (non-pungent) capsiate reduced food intake by similar degree (Kwon et al. J. Nutr. Biochem. 24:1078, 2013). All things considered, I do not believe that aversion to capsaicin explains the effects summarized in Table 1.

Thank you for pointing out serum albumin. Indeed, binding to serum albumin should further reduce the concentration of bioavailable capsaicin. This is now mentioned in the text (with new reference 180). 

Reviewer 3 Report

Dear Author

I read your very interesting review on spicy food and effects of capsaicine on componnets of metabolic syndrome and underlying mechanism of action in experimental and clinical settings with true satisfaction. Your review covers a large area of research and many multidisciplinary aspects, but is still easy to read and in my opinion has an important didactic value.

I have only a few minor remarks.

1 to the text:

Page 2 Line 91: “These changes were accompanied by improved total-cholesterol levels … “the word improved can be misunderstood in this context maybe is better to use decreased

Page 2 Line 93 “ameliorated the rise in serum lipids…” same comment as above

Page 3 Line 132 “ …that the mortality rate among chili-eaters (21.6%) was statistically superior to that (33.6%) of the non-eaters.” Maybe for unequivocal understanding : statistically significantly lower

Page 9 Line 395 ‘Dietary capsaicin was shown to ameliorate the development of atherosclerotic plaques’ Could you change the wording?

2. To Figure and  tables

You mentioned in Figure 1 that it is reproduced with permission of the publisher.

 In your review, you also present table 1 (data from reference 38) and table 2 (data from reference 43). Also, note that those tables are reproduced with permission.

Author Response

Dear Referee: thank you for your suggestions that I fully adopted (for easier recognition, changes are highlighted in yellow in the revised MS).

Lines 91, 93, 132, and 395: the wording has been changed as you suggested.

Table 1 and 2 are not reproductions, these are my creations based on data published in reference 38 and 43.